

# Contribution of storms to shoreline changes in mesotidal dissipative beaches. Case study in the Gulf of Cadiz (SW Spain)

María Puig[1], Laura Del Río[1], Theocharis A. Plomaritis[1,2], Javier Benavente[1]

[1]Dpt. Earth Science, CASEM, University of Cádiz, Av. República Saharaui s/n  Puerto Real, Cádiz, 11510, Spain.
[2]CIMA, University of Algarve, Campus of Gambelas, Faro, 8005-135, Portugal.

*Correspondence to:* María Puig (maria.puig@uca.es)

**Abstract**. In this study an analysis of storminess and rates of shoreline change is performed and discussed jointly in four geomorphological units of the Gulf of Cadiz (SW Spain), for the period 1956-2010. For this purpose, storm events are identified based on the following characteristics: wave height above 2.5 m, a minimum duration of 12 h and events with calm periods of less than 24 h were considered as a single event. Subsequently, energy parameters are determined in order to characterize storm-induced impacts. On the other hand, GIS tools are used to measure shoreline changes on aerial photographs and orthophotographs of each site, selecting the high water line as shoreline proxy. Each geomorphological unit is divided into different behavioural patterns according to recorded coastal changes, so that each one shows a particular behaviour.

In general the variability of shoreline changes that is explained by storms and the relation between storm parameters and coastal changes present better results in exposed areas (Cadiz and Vistahermosa) than in sheltered areas (Valdelagrana spit barrier), as the former are more sensitive to storm impacts. On the contrary, in areas where there is no relation between coastal changes and storm parameters (Valdelagrana and Sancti Petri sand spit), it is suggested that anthropogenic factors are the main forcing agents determining shoreline behaviour. However, in these areas the storminess also modulates coastline recession by increasing erosion when the number of storms is high.

## 1 Introduction

Accelerating population growth in coastal areas and the threat of climate change have greatly increased the interest of researchers in the dynamic behaviour of the shoreline (Barragán and De Andrés, 2015). Beside scientists, coastal managers also have the need to know how shoreline position changes over time in order to develop adequate coastal planning policies. Coastal areas are subject to a variety of hydrodynamic and geomorphological processes that occur over the short, medium and long time-scales. Whether natural or man-induced, coasts all over the world are affected by a continuous balance between erosion-accretion processes. The contribution of different factors to the above processes in the medium term has been studied by numerous authors. On one hand, within natural controls, the main contributing agents are geological framework, sediment supply and wave climate (e.g. Jackson et al., 2005; Dissanayake et al., 2015; Masselink et al., 2016).



Within anthropogenic controls, land use transformation, coastal development and the construction of engineering structures at the coast (such as jetties) and in river basins (dams) are the major causes of shoreline erosion (e.g. Di Stefano et al., 2013; Hapke et al., 2013; Van Wesenbeeck et al., 2015).

Regarding the influence of storms in coastal changes, shoreline response in sandy beaches depends mainly on the characteristics of the storm, sediment supply and nearshore bathymetry. The most important parameters of a storm related to the potential generation of morphological changes are wave characteristics, storm duration and clustering and storm track (Morton, 2002; Bertin, 2012). Ferreira (2005) showed that storm groups with low return periods can cause the same erosion as a single extreme storm with high return period. In this respect, Plomaritis et al., (2015) showed that in the Gulf of Cadiz (SW Spain) the number of storms and their duration are related with large scale atmospheric circulation patterns, namely the NAO and the EA, and that larger number of storms increase the probabilities of events with higher return period. Based on the above controls, storms produce numerous effects that include beach and dune erosion, coastal flooding, inlet breaching and overwash. In economic terms, storm impacts can be significant and in some cases they can have devastating consequences (Ciavola et al., 2011; Kreibich et al., 2014). Thus, the development of storm hazard and vulnerability assessmentsis an issue of primary concern in coastal zones.

Aerial photographs and ortophotographs provide snapshots of coastal morphology that can give an overview of the above mentioned storm effects, and they are the most commonly used technique to assess medium- and long-term shoreline evolution. Furthermore, they are the basis of numerous models that have been developed in order to predict future shoreline changes (e.g. Cowell et al., 1995; Frazer et al., 2009; Anderson et al., 2010). In the Gulf of Cadiz, shoreline changes over the last 50 years have been studied by several authors (Benavente et al., 2006; Anfuso et al., 2007; Del Río et al., 2013). According to these studies the area exhibits overall erosional trends; however, there are particular cases of rectilinear beaches with stable or accreting patterns.

The relation between storminess and morphological response in the short term has also been widely studied in the Gulf of Cadiz (e.g. Reyes et al., 1999; Rangel-Buitrago and Anfuso, 2011a). However, to date there is a lack of detailed studies on the contribution of storms to coastal evolution over the medium-term scale (a few decades) and on a regional scale application. This information is of great interest, given the existing uncertainty about how climate variability can induce variations in storm patterns.It can also be useful for erosion management purposes, since in the last two decades, in order to mitigate coastal erosion impacts a total of 90 beach nourishments have been performed on the sandy shores of Cadiz province (Muñoz-Perez et al., 2001, 2014).

Within this context, in the present work a comparison between medium term shoreline changes and wave climate data is undertaken in four geomorphological units located in the southern Gulf of Cadiz, and the influence of storminess in coastal evolution is analysed and discussed. For this purpose, storm record is obtained by combining modelled data of a hindcast database and measured data. Then, shoreline changes are obtained by analysing sets of aerial photographs and ortophotographs dating between 1956 and 2010. Finally, results on shoreline changes and storm record, as well as the relation between them in the study area, are presented and discussed.



## 2 Study area

The study area is constituted by four geomorphological units located in the southern Gulf of Cadiz (SW Spain) (Fig. 1) with contrasting geomorphological characteristics, degree of urban development and exposure to waves (Table 1): Vistahermosa (I), Valdelagrana spit barrier (II), Cadiz (III) and Sancti Petri sand spit (IV).

Vistahermosa and Valdelagrana geomorphological units are located in the Northern sector of the Bay of Cadiz and present a restriction in the sediment budget as they are limited by headlands and engineering structures (Benavente et al., 2006) (Fig. 1). Vistahermosa unit is a Z-bay (crenulated shaped beach) that extends along 5 km covering three sandy beaches from North to South: El Almirante, Fuentebravía and Santa Catalina beaches. The northern part, which presents WNW-ESE orientation, is protected from storm waves that approach mainly from western-southwestern directions. Conversely, the rest of the unit due to its coastline orientation (NNW-SSE) is exposed to storm waves. It is backed by artificially stabilized sandy cliffs of increasing height to the North (up to 25 m), and although there is a semi-natural zone in its central part, over the last decades the entire beach has been heavily transformed for tourism purposes and erosion control (rip-rap revetments and seawalls). Valdelagrana geomorphological unit, with a total length of 7 km and located in the inner part of the Bay, is a well-developed spit barrier that has recently evolved into a Z-bay due to the installation and extension of the Guadalete river jetties at its north end (Martínez-del-Pozo et al., 2001). With a general N-S orientation, it is sheltered from wave incidence, as waves approaching from W and SW are refracted and diffracted around Cadiz tombolo (Fig. 1). The northern part is heavily urbanized, while the central and southern sectors belong to the Cadiz Bay Natural Park.

The Southern units (Cadiz and Sancti Petri sand spit), with similar orientation (NNW-SSE) are situated in the outer part of the Bay, thus being exposed to direct wave impact (Fig. 1). Both units are constituted by long and rectilinear sandy beaches and their sediment budget is connected as prevailing longshore transport drifts southward. The geomorphological unit of Cadiz, with a length of 9 km, is constituted by urban and natural areas (Fig. 1). The northern beaches, which are backed by a seawall, belong to the city of Cadiz, while the southernmost beaches are in a natural area backed by low foredunes. Finally, Sancti Petri sand spit is a natural zone located in the southern part of the Bay of Cadiz. It extends along 7.7 km, being also part of Cadiz Bay Natural Park. This sandy unit is backed by dune ridges and extensive salt marshes, and limited inland by Sancti Petri tidal channel.

In the Bay of Cadiz subtidal and intertidal rocky shore platforms are present along the coast. On one hand, subtidal rocky platforms appear discontinuously along the outer part of the Bay (Cadiz and Camposoto). On the other hand, intertidal



platforms are present at several points along Vistahermosa, as well as in the northern sector and the natural part of Cadiz unit (Fig.1).

The area is mesotidal and semidiurnal with mean neap and spring tidal ranges between 1.20 to 2.96 m respectively (Instituto Hidrográfico de la Marina, 2014). Dominant winds in the Gulf of Cadiz blow from W-SW and E-SE directions. ESE winds come from the Mediterranean Sea and have low influence on coastal erosion due to their short fetch and the coastline orientation. On the contrary, WSW winds with longer fetch have greater importance (Gracia et al., 2006), especially during storm periods, which are considered from November to March (Puertos del Estado, 2006). Dominant waves approach from the West (Fig. 2), leading to the above mentioned southward direction of longshore drift in the area. The average significant wave height is about 1 m with associated periods of 5-6 seconds, hence Cadiz littoral is considered as a low energy coast according to the classification by Davis and Hayes (1984). In this respect, different thresholds have been proposed for storm waves in the coast of Cadiz. On one hand, Del Río et al. (2012) have suggested a minimum wave height threshold between 1 and 3.75 m to generate morphological changes on beaches along this coast, and between 3.3 and 3.75 m to produce dune foot erosion. On the other hand, analysing oceanographic conditions Ribera et al. (2011) have proposed a minimum threshold for storm waves between 2.2 and 2.5 m, after correlating historical and instrumental storm series. Wave pattern is characterized by a clear high energy-winter/low energy-summer duality; however, there are mild winter seasons that generate intensive beach recoveries (Benavente et al., 2014) due to the above mentioned low-energy character of the Cadiz coast.

## 3 Methods

### 3.1 Wave data and storm selection

A synthetic storm record was constructed by combining wave data from the coastal wave buoy of Cadiz (Fig. 1) and those from the hindcast database of the HIPOCAS project (Guedes-Soares et al., 2002). Inter-calibration of the two records was undertaken for the overlapping period with directional data (year 2001) by using a peak over threshold analysis (POT). For this purpose, the approach by Plomaritis et al. (2015) was followed emphasizing on agreement during storms.

The storm events were identified using again POT analysis following the criteria described in Del Río et al. (2012), i.e. wave heights over 2.5 m and a minimum duration of 12 hours; storm groups with calm periods of less than 24 hours between them were considered a single storm-group event. The above threshold for significant wave height coincides with the 0.95 quantile defined by Masselink et al. (2014) and with the minimum wave height threshold proposed by Ribera et al. (2011).





## 3.2 Storm characterization

Storm-induced impacts at the shoreline were characterized by computing different parameters, namely wave energy ($E$), wave energy at high tide, wave power ($P$), wave erosivity ($Er$), number of storms, and storm duration for each storm season. A wide variety of behaviours for each parameter were covered, by calculating the mean, cumulative and peak of each

parameter. The rationale is that cumulative values include variations in storm frequency and storm duration, while mean values reflect an average value that compresses the storm variability, and peak values show the intensity of the strongest storm during the season.

In order to calculate wave characteristics close to the shoreline and account for the different coastline orientation of the four geomorphological units described above, wave transformation due to shoaling and refraction was undertaken by using a

linear wave theory (Kamphuis, 2000):

$$H_{s,b} = K_s K_r H_{o,s} \tag{1}$$

where $H_{s,b}$ is the significant wave height at breaking, $K_s$ is shoaling coefficient, $K_r$ is refraction coefficient and $H_{o,s}$ is wave height at the wave buoy depth (21 m).

Wave energy at breaking ($E$) was derived from the equation proposed by Dean and Dalrymple (1991):

$$E = 1/16 \rho g\, H_{s,b}^2 \tag{2}$$

Where $\rho$ is water density and $g$ is gravitacional constant. Subsequently, wave energy at the time of high tide was obtained in order to account for the importance of the tide during storms, as previous studies in the area have shown that the largest contribution to total sea level variation is the tide (Del Río et al., 2012). In the same way, wave power or wave energy flux at breaking was calculated by:

$$P = E\, C_g \tag{3}$$

In this expression $C_g = 0.5(gH_{s,b}/\gamma)$ is the shallow water group velocity, where $\gamma$ is the breaking parameter. In the present work, McCowan's (1894) breaking criterion was used ($\gamma=0.78$). Finally, wave erosivity (Benavente et al., 2000) was introduced to distinguish erosive conditions from accreting conditions. This parameter is an indicative of the erosive potential of incident waves and is given by the product of the dimensionless fall velocity parameter ($\Omega$) and wave energy at

breaking ($E$):

$$\Omega = H_{s,b} / (w T_p) \tag{4}$$

$$E_r = E\Omega = k_d H_{s,b} / T_p \tag{5}$$

Where $T_p$ is peak wave period and $k_d = \rho g/16w$ is a constant that includes the sediment fall velocity ($w$), which is function of the median grain size (D50) and the density of sand. For the study site a density of 2.65 was considered (density of quartz).



### 3.3 Shoreline changes

For each geomorphological unit an average of 15 aerial photographs and orthophotographs were used at scales ranging from 1:15,000 to 1:33,000 in order to measure changes in shoreline position over a 54-year span (1956-2010) (Table 2). Erosion-accretion rates were calculated using the methods described in detail in Del Río et al. (2013), by employing GIS tools for the

georeferencing of the aerial photographs, digitization of shoreline proxies, calculation of shoreline changes and determination of associated uncertainty (Del Río and Gracia, 2013). Regarding shoreline indicators, in order to use a common proxy in all the sites the high water line (considered as the location of the wet/dry beach contact) was used as shoreline indicator. This proxy was also used in the cliffs of northen Vistahermosa , as it coincides with the cliff foot. It was not possible to use the dune foot as a common proxy, due to the absence of dunes at the backbeach in the study sites of Cadiz

and Vistahermosa.

After shoreline digitization, the ArcGIS™ extension Digital Shoreline Analysis System (DSAS) (Thieler et al., 2009) was

used to measure the distance between shorelines along shore-normal transects spaced at 20 m intervals. Based on this, rates of shoreline change on each transect were obtained by dividing the distance between consecutive shorelines by the time interval between them.

It must be noted that calculated shoreline changes were influenced by the time interval between photographs, which tends to generate overestimated rates of shoreline change during particularly short time spans and underestimated rates during very

long time spans (Dolan et al., 1991). This error is acknowledged here but its analysis is out of the scope of this work.

### 3.4 Relation between storm parameters and shoreline changes

Summarizing the above described method, a total of 6 storm variables (energy, energy at high tide, wave power, erosivity and storm frequency and duration) were selected in order to characterize shoreline changes. From the first four parameters, their mean, cumulative and peak were calculated giving a total of 14 variables.

Storm parameters were computed for the periods between photographs and their contribution to shoreline changes was analysed in two different ways. Firstly, the relation between storm parameters and rates of shoreline change was assessed without considering coastal exposure and storm distribution along the photographs. Secondly, coastal exposure was considered (Eq. 1) in calculating the storm parameters. In addition to this, a weighting factor was assigned, which increased linearly with the temporal proximity to the photographs. The weighting factor assumes a reduced impact of the older storms

in coastline retreat and is similar to the recovery rate approach by Frazer et al. (2009). The zero weight was given to the storm parameters dating more than five years before the photograph, as in the study area, beaches require several years to





recover from the impacts of strongest storms in the medium term (Benavente et al., 2013). Moreover, this period coincides with the mean sampling interval between consecutive photographs.

The relation between storm parameters and rates of shoreline change was assessed by means of the Pearson correlation coefficient (R), and the p-value of <0.05 was used to check the statistical significance of the correlation. Furthermore, for the

5 cases of strong relationship between storm parameters and erosion rates a linear and nonlinear fit was used and the $R^2$ was calculated, in order to estimate the explained variability between the storm parameters and the shoreline evolution.

## 4 Results

### 4.1 Storm analysis

The effects of wave transformation due to shoaling and refraction, and the temporal evolution of energetic parameters are

10 shown in Fig. 3. It presents 4 of the 6 storm parameters that were analysed in the present study (energy, energy at high tide, number of storms and duration) for Cadiz unit. The other parameters (wave power and wave erosivity factor) and the rest of areas are not shown as they present very similar patterns.

The effect on coastal orientation, when wave transformation is considered, does not produce large variations in the patterns of storm parameters; however, it increases their values along the studied period (Fig. 3a and 3b). Energy (Fig. 3a) (and similarly wave power and erosivity) is in both cases characterized by a relatively stable pattern from 1956 to 1995; followed by a negative trend until 2004 and an increase over the last years. Conversely, energy during high tide (Fig. 3b) has a large

interannual variability with several remarkable peaks (i.e. 1961, 1981,1987, 1992, 1995, 1998 and 2009).

Over the study period a total of 231 storms were recorded, with an overall duration of 351 days (Fig. 3c and 3d). The stormiest years, with more than 8 storms per year and over 100 days of duration, are 1958, 1963, 1996, 2003, 2009 and 2010. Nevertheless, there are other years (i.e. 1979) where the number of storms is lower than 8 but with over 100 days of duration. Years with low storm record (less than two storms per year and 15 days of duration) are 1971, 1974, 1980, 1988, 1990, 1992,

1993 and 2007. Finally, the remaining periods present an average of four storms per year and 45 days of duration.

Regarding the values of the above parameters calculated for the 14 periods between photographs (Fig. 4), storm duration and frequency and cumulative and mean values of parameters present similar pattern. Conversely, peak values of each parameter differ from the above trends showing a more variable pattern (Fig. 4c and 4d). Nevertheless some peaks, i.e. 1984-1985 and 2002-2005, match with the occurrences of the highest cumulative parameters.



## 4.2 Shoreline changes

In general, shoreline changes along the study zone show a high spatial and temporal variability. The four geomorphological units present different behaviours and within each unit, there are variable patterns.

Shoreline trends along the four study sites between 1956 and 2010 are shown in Table 3 and in Fig.5 and 6. Each one of the

5    four units has been divided into behavioural patterns (BP hereafter) according to the general shape of the fitting curve of the shoreline trend. In the following paragraphs each BP is explained according to their mean shoreline position and rates along the period of study.

The first two behavioural patterns of Vistahermosa involve an erosional trend, however they are separated due to the availability of aerial photographs and the alongshore variability (Fig. 5a). In detail, the northern part (BP1), with an average retreat rate of -1.1 m/yr, shows continuous erosion until 2005, followed by a stabilization. Further South, in Fuentebravía and in the southern part of Santa Catalina beach (corresponding to the central and southern portions of BP2), a comparable

average retreat is recorded (-0.4 m/yr). On the other hand, BP3, situated in the central part of Santa Catalina beach, presents the most variable trend with mean accretion rates ranging from 2.3 to 9.8 m/yr and mean erosional rates from -0.9 to -15.8 m/yr. Nevertheless, these extreme rates (-15.8 m/yr and 9.8 m/yr) coincide with particularly short sampling intervals (1984-1985 and 1998-2000) and could thus be affected by the above mentioned overestimation.

The northern part of Valdelagrana spit barrier (BP 4) is characterized by a general accretionary trend, with shift to erosion

over the last years (Fig. 5b). The central part of the spit (BP5) experiences similar trends between 1956 and 1984, and then an erosional pattern until 1994. This is followed by accretion until 2002 and a gradual shoreline retreat over the last decade. The southern sector of Valdelagrana (BP 6) shows an extremely eroding trend almost during the entire period. This extreme shoreline recession appears between 1976 and 1977 (-31.4 m/yr) after the construction of the jetties in the Guadalete river mouth (Martinez-del-Pozo et al., 2001), and it slows down during 1998-2000 and 2005-2007 reaching values of -0.5 m/yr.

Over the last years, the erosion rate increases again to -12.2 m/yr.

The urban part of Cadiz unit (BP7) shows a slightly erosional trend from 1956 to 1985 (Fig. 6a). This is followed by an

accretionary period (12 m/yr) from 1985-1992 and in the last decades, this area experiences gradual erosion. Southwards, the natural section of the unit (BP8) presents a roughly stable net balance considering the shoreline position of the first (1956) and the last photograph (2010). It shows a progressive accretion, with a mean rate of 5.1 m/yr, after an erosional behaviour (-1.2 m/yr), that occurs during the first period between photos (1956-1977).



Finally, Sancti Petri experiences a continuous erosion (BP9) along its coast with an average retreat rate of -1.7 m/yr, which includes erosion rates from -1.8 to -9.2 and accretion rates between 0.3 and 17.4 m/yr (Fig. 6b).

## 4.3 Relation between shoreline changes and storm parameters

The relation between shoreline changes and storm parameters, without taking into account the wave transformation and weighting, shows low correlations (r<0.5) for all the geomorphological units, excepting the exposed unit of Cadiz (BP8), where the storm duration and the cumulative parameters of energy, wave power and erosivity present significant correlations (r=-0.57, r=-0.56, r=-0.53 and r=-0.55 respectively).

Consideration of wave transformation and the storm weighting involves an improvement of the statistical results (Table 4). Within this analysis, linear fitting yields better results in all study sites than nonlinear fitting, except in the central part of Valdelagrana (BP5), where the exponential correlation between shoreline changes and storm frequency shows better values (r=-0.60, p=0.01) than linear correlation (r=-0.44, p=0.09).

Analysing the correlation of storm parameters (Table 4), energy at high tide is with the one showing the highest values, with significant correlations in five of the BPs (BP2, BP3, BP4, BP7 and BP8). This is followed by wave power and energy, which present significant correlations with shoreline changes in four BPs (BP3, BP4, BP7 and BP8). Erosivity and storm duration show good results with three BPs (BP4, BP7 and BP8), and finally, storm frequency is not correlated with any of the BPs. On the other hand, the correlated variables are mostly cumulative parameters, followed by peak parameters and mean parameters.

On a site-by-site basis, statistical results in Vistahermosa show variable values along the unit, with the best connection appearing in the sandy beaches of BP2 with cumulative and peak energy at high tide (Fig. 7a) and in BP3 with cumulative and peakenergy, cumulative and peak wave power, and peak energy at high tide (Table 4). However, the variability related to storminess is low, as none of these parameters exceed 50%. On the contrary, shoreline changes in BP1 are not correlated with any of the storm parameters.

Valdelagrana is characterized by showing only strong correlations at its northern part (BP4) with almost all storm parameters (Table 4), namely the mean and cumulative of energy, wave power, erosivity and energy at high tide, the peak of energy at high tide and storm duration (Fig. 7b). Within these parameters, peak energy at high tide explains better the variability of shoreline changes (48%) than the rest of parameters. Moreover, if periods with intensive shoreline accretion and low energy are removed from the record, the percentage of shoreline changes explained by peak energy at high tide increases to 58%.



Regarding Cadiz area, rates of shoreline change at the urban part (BP7) are better correlated with the cumulative of energy, wave power, erosivity and energy at high tide, and with storm duration (Table 4), while the natural part of the unit (BP8) presents significant correlation with almost all parameters excepting the peak of wave power, the cumulative of energy at high tide and storm frequency (Fig. 7c). The variation of shoreline changes that is explained by the correlated parameters does not exceed 35%; however, once again if periods with low energy and shoreline accretion are extracted from the dataset, almost 50% of shoreline changes are explained by the following storm variables: storm duration (47%) and cumulative erosivity (40%). The rest of correlated parameters increase slightly but explain less than 40% of shoreline changes.

Finally, storm parameters at Sancti Petri sand spits (BP) explain a small percentage of shoreline variability, as the correlations are very poor (Table 4), with none of the parameters being statistically significant along the section (Fig. 7d).

## 5 Discussion

### 5.1 Storm record and shoreline changes

When comparing the results obtained in this work with literature on storms in the study area, the temporal pattern of storm energy and the number of storms (Fig. 3) coincide with the periods of negative values of NAO index described in Rangel-Buitrago and Anfuso (2011b) for the Gulf of Cadiz, thus confirming the relation observed by Plomaritis et al. (2015) between storm record and negative NAO in this area. However, the annual energy at high tide shows a different pattern, which is not related to NAO index. In fact, this parameter is mainly related to the tide state rather than to wave height.

Storm record along the periods between photographs (Fig. 4) also matches the years of high Storm Power Index obtained by Rangel-Buitrago and Anfuso (2011b) in the Bay of Cadiz and by Almeida et al. (2011), who analyse storminess in Faro (Portugal), located in the western Gulf of Cadiz. Nevertheless, other years with high storm record according to these studies (1987 and 1989), which correspond to the sampling period 1985-1992 in the present work, are not reflected in the obtained results. These may be related to the fact that other years included in that period have a low Storm Power Index (i.e. 1988, 1990, 1991 and 1992) (Rangel-Buitrago and Anfuso, 2011b).

Regarding rates of shoreline change, the present work evaluates new datasets recording a wide variety of shoreline trends along the Bay of Cadiz, and confirms the results of previous works. Significant accretionary and erosional rates have previously been reported in the northern and southern sections of Valdelagrana respectively (Benavente et al., 2006), due to the construction of jetties in the mouth of Guadalete river and subsequent changes in the diffraction control point of the headland-bay system. Other patterns, such as the stable trend along the natural part of Cadiz and the erosional rates of Fuentebravía and Sancti Petri, have also been mentioned in previous works (Del Río et al., 2012).





## 5.2 Correlation analysis

Storm influence on coastal changes has been largely describedin the literature (e.g. Houser et al., 2008; Castelle et al., 2015). The present work analyses this effect indifferent areas of the Bay of Cadiz. In general, there are better correlations between coastline changes and storm parameters in exposed areas (BP 2 and BP3 of Vistahermosa and BP7 and BP8 of Cadiz) than in sheltered areas (BP5 and BP6 of Valdelagrana spit barrier). However, Sancti Petri sand spit and the northern part of Valdelagrana (BP4) constitute an exception to this, as the first one is a exposed unit and presents low correlation values, while the second one is a sheltered sector and shows strong correlations between shoreline changes and storm parameters (Table 4). In the areas where correlations are significant, the effects of severe storms in sediment budget can be mainly related to the offshore transport of beach sediments, which are during the following years mobilized back to the beach by fair weather waves (Benavente et al., 2013).

In general, storm parameters explain a negligible or small part of the system variability. Nevertheless, almost half of the variability of some sections, i.e. BP4 of Valdelagrana and BP7 of Cadiz, is explained by storms. Moreover, this proportion is increased when periods with intensive shoreline accretion and low energy are removed from the statistical analysis. The shoreline rates of these periods are related with human interventions in some areas, such as the massive nourishment that was carried out in 1991 in Cadiz urban beach (Muñoz-Pérez et al., 2001), that could be responsible for the intensive accretion recorded between 1986 and 1992 in BP8.

The correlated variables are mostly cumulative and peak parameters, especially the energy and energy at high tide. This is due to the fact that in dissipative beach systems, like the ones described here, the total erosion is frequently associated with the largest storms, because the recovery processes are slow and in general there is very little inter-storm recovery (Coco et al., 2014). Furthermore, it must be taken into consideration that in most cases, only the extreme events (that are represented in cumulative and peak parameters) are capable of promoting coastline retreat that persists for a duration of time long enough to be captured by the temporal resolution of aerial photos. When tide state is considered (i.e. parameters at high tide), the correlation values improve. This agrees with Del Río et al. (2012), who highlighted the importance of tide-related sea level variations in the Bay of Cadiz, where the percentage of surge variability is only a small fraction of the tidal variability, which actually controls the position of the waterline during storm events. In addition to this, in dissipative coasts as the study zone, with a relative tidal range (RTR) of 3 and dimensionless fall velocity parameter ($\Omega$) of 5.4, surf and swash processes, that promote erosion during storms, are dominant only close to the spring high water mark (Masselink and Short, 1993).

The fact that shoreline changes correlate with a specific parameter on each site shows that there is no relation between parameters and sites characteristics. For instance, the urban unit of Vistahermosa (BP2) and the natural part of Cadiz (BP7), which have similar wave exposure, exhibit meaningful correlations with different parameters (Table 4), meaning that the behaviour of each unit is influenced by local factors.

Regarding correlation results (Table 4) and analysing each unit one by one, the poor correlation of the northern part of Vistahermosa (El Almirante beach) (BP1) could be explained by geological setting, as cliffs are constituted by soft sands and

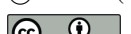



marls. This vulnerable material is affected by hydric erosion and landslide processes (Del Río et al., 2009), so cliff retreat is continuous after each storm event, even during low-energy periods. On the other hand, the strong correlation obtained in BP2 and BP3 could be expected, as this area is highly exposed to wave attack. However, storms explain only between 30 and 38% of the shoreline variability in these areas and may be because in the northern part of BP 2 (Fuentebravia beach) storm contribution is over imposed on the effects of the harbour at Rota NATO Base (located immediately NW of the unit, see Fig. 1), which interrupts sediment transport southward and generates strong downdrift erosion (Cooper et al., 2009). Consequently, in this beach nourishment works have repeatedly been carried out between 1992 and 2010, with a total of 12 replenishments (Muñoz-Pérez et al., 2001, 2014).

In the central and southern sectors of Valelagrana (BP5 and BP6), the low percentage of coastline variability explained by storms and the low correlation values between shoreline changes and storm parameters suggest that, besides being protected from wave incidence, there are other factors affecting shoreline changes along the analysed periods. As explained above, the most important interventions having triggered significant changes in this area include the building of jetties in Guadalete river mouth, having led to extreme erosion in the southernmost sector of Valdelagrana (Martinez-del-Pozo et al., 2001). Nevertheless, a certain correlation was expected in the BP5 of the unit, as it is the most exposed sector of Valdelagrana to wave attack and it is situated in the central part of the Z-bay (pivoting zone), where the lowest shoreline variation to rotational movement occurs and the net longshore transport is close to zero (Short and Masselink, 2001). Conversely, the strong correlation observed in the northern part of this area (BP4) indicates that this sector is more susceptible to storms than to the accretionary effects of the aforementioned jetties.

In the urban part of Cadiz (BP7), several factors must be taken into account to explain the low percentage of variability explained by storms. First, two groynes were constructed in the northern and southern limits of Santa Maria Beach (Fig. 6) in the 1980s and lengthened in the 1990s, in order to counteract beach sand loss. Later on (1997-1998) a submerged breakwater was built at a depth of 3 meters in front of Santa Maria Beach. Finally, a total of 7 nourishments have been performed between 1991 and 2010 along the whole urban beach (Muñoz-Perez et al., 2001, 2014). All these interventions have had an important impact on shoreline evolution regardless of storm events. As for the natural part of Cadiz unit (BP8), it presents strong correlations with peak parameters and parameters that depend on storm duration (i.e. cumulative variables), so there is a clear relation between storminess and shoreline changes. However, due to the proximity to the urban area located updrift, other factors could contribute to shoreline change, especially the aforementioned massive nourishment carried out in the urban beach in 1991, with a total volume of 2,000,000 m$^3$ (Muñoz-Pérez et al., 2001). In fact, if the period containing this intervention (1986-1992) is removed from the analysis, half of the shoreline variability is explained by storms.

The absence of correlation observed between shoreline changes and storm parameters in BP9 of Sancti Petri agrees with the recent findings by Benavente et al. (2013). These authors showed the vulnerability of this area to wave erosion during low energy conditions, and the storm thresholds used in the present work, which is the minimum wave height that Ribera et al. (2011) assumed as adequate, exclude erosion events during low energy conditions.



It is clear from the above results that, apart from the relationship between storminess and shoreline change, shoreline position in the study area can be affected by other natural factors, such as coastal bathymetry and orientation, geological framework and hydrodynamic conditions. According to Del Río et al. (2013), in the external Bay of Cadiz the presence of subtidal rocky shore platforms along the coast plays an important role influencing patterns of shoreline change (Fig. 1). Depending on their length and location, they produce erosion-accretion processes as they modify diffraction and refraction wave patterns. The effects of erosion could be shown in Sancti Petri sand spit, in the northern part of BP9, because the headland at its northernmost end (Fig. 1, Fig. 6b) would act as a natural groyne obstructing sand movement. On the other hand, there are other areas that despite presenting rocky shore platforms, have not recorded a clear erosion pattern. This occurs in the natural part of Cadiz (BP8), where the largest platform of the study area appears (Fig. 1). A reason for this could be that the gaps observed in the rocky platform cause a variable behaviour, generating erosion where wave energy is concentrated and accumulation in the shadow areas. This effect is visible on the coastal morphology of BP8, such as the embayments of El Caido and La Leona (accretion/recession)(Fig.6a).

On the contrary, stabilization or accumulation should be recorded next to Sancti Petri tidal inlet (BP9) and in the northern and southern ends of Vistahermosa (BP 1 and 2), as in both cases the platform located downdrift could act as a groyne capturing sand from longshore drift. Nevertheless, both areas show an erosion pattern in the medium-term behaviour. In Vistahermosa this trend could be related to the sediment deficit provoked by the aforementioned jetties at Rota NATO Base, which in the northern area (El Almirante beach) would also be added to the above mentioned weakness of the cliffs. On the other hand, the erosion of the southern part of Sancti Petri (BP9) could be related to the dynamics of a sandy shoal that appears at the south of Punta del Boqueron due to the longshore sediment transport and the complex tidal currents existing in this area (Del Río et al., 2013).

## 5.3 Considerations on the methodology

The methods used in the present work give a sense of how storm parameters contribute to shoreline evolution in the Gulf of Cadiz in the period of study. The success of shoreline change analysis depends fundamentally on the accuracy of photographs and the precision in measuring coastal changes (Moore, 2000). Image quality can also influence shoreline position, increasing alongshore variability shown by error bars in Figs. 5 and 6. On the other hand, selected shoreline proxies can be a source of error, as the position of the high water line is affected by meteorological (wind and wave) conditions, tidal effects and beach seasonality. The latter is especially significant for the period between 1956-1976, because the first aerial photographs of the record (1956) were taken in winter; this way, sections with strong seasonal differences in beach slope (Cadiz, Sancti Petri sand spit and Vistahermosa) (Del Río et al., 2013) have variable positions of the high water line. The remaining shorelines correspond to photographs taken during the summer, thus avoiding beach seasonality error. In addition to this, artificial nourishments carried out in the area also affect the position of the high water line. The most distinctive example is the above mentioned massive replenishment performed in the urban beach of Cadiz, which caused a shoreline advance of approximately 80 m during 1990s. Nevertheless and despite the above considerations, due to the need for a



common shoreline proxy between the different study sites, the high water line was accepted as representative of shoreline position, in accordance with authors such as Crowell et al. (1997) and Gorman et al. (1998).

On the other hand, as previously mentioned, the sampling interval of photographs can increase uncertainty associated to this methodology. Short periods between photographs can produce overestimation of the rates of shoreline change (e.g. the accretion rate of 9.8 m/yr observed in BP3 of Vistahermosa between 1998-2000, Fig. 5a). Conversely, long periods between photographs tend to smooth the variability of changes (Dolan et al., 1991), thus yielding lower rates (e.g. the erosion rate of-0.7 m/yr observed in BP7 of Cadiz between 1956-1976 (Fig. 6a).

Finally, it is important to remark that aerial photographs represent snapshots of the coastline, thus, storms occurring shortly before the photographs should have more impact on the coastline. This fact has been accounted in the weighting procedure done in the present work, where a linear distinction of 5 years has been considered, based on the temporal proximity of the storms to the photographs. Although there are slight differences on post-storm recovery processes between intermediate and dissipative beaches in the Bay of Cadiz, beach recovery take years to occur (Benavente et al., 2013). Thus, this period of 5 years has been considered as representative in this respect. Moreover, as previously mentioned, it is the mean sampling interval between consecutive photographs. In this sense, the weighting procedure has remarkably modified the results of correlations in long intervals (i.e. 1956-1976) while having a negligible effect during short intervals (i.e. 2000-2002 and 2005-2007).

On that note, Davidson et al. (2013) proposed a response factor of the beach as an implicit free parameter of a model that can predict shoreline changes in the short and medium term. In the future, further investigation will be done with this model in order to calculate the response factor parameter in each study site and introduce it as a weighting factor on the storms.

## 6 Conclusions

In this work four beaches of the Bay of Cadiz with diverse characteristics of wave exposure and human development were analysed in order to assess the contribution of storms in recorded shoreline changes. In general, it has been found that although there is a correlation between shoreline changes and storm variables, the latter explain a low percentage of this variability as in most of the sites $R^2$ is lower than 0.5. However, it has been found that in some areas when periods with intensive shoreline accretion and low energy are removed from the statistical analysis, their system variability is explained by storminess (i.e. northern part of Valdelagrana spit barrier and southern part of Cadiz). These periods coincide in the case of Cadiz with years when human interventions significantly altered natural shoreline behaviour.

The energetic parameters of the storms were found to be more correlated with shoreline changes at exposed areas than at embayed beaches. Tide consideration was decisive as statistical results improved considerably. On the other hand, these storm-related rates of shoreline change were specific to a particular parameter on each site, so each area is suggested to be influenced by local behaviour. The most strongly correlated parameters were cumulative parameters (cumulative energy, wave power, erosivity and energy at high tide) and peak parameters (cumulative energy, erosivity and energy at high tide).



Finally, it is suggested that anthropogenic factors are the main forcing agents determining shoreline behaviour in the study area, together with other natural factors, such as geological framework, which controls coastal bathymetry and orientation (e.g. the distribution of rocky shore platforms and sandy shoals along the study zone). However, even in the absence of correlation between coastal changes and storm parameters, the storminess contributes to modulate shoreline recession, with increasing erosion during periods when the number of storms is higher.

This work highlights that in geomorphologically complex areas with variable type of uses and management plans, the response of the coast to storm events and patterns is not uniform. Selected areas with different characteristics cover a wide range of behaviours and a different response (more or less sensitive) to storm events. This relation between storminess and shoreline change can contribute to the prediction of short- and medium-term coastal variations. In this respect, further research will be focused on the analysis of short-term beach evolution, in order to assess the response rate of each site to specific storm events and compare it with the results obtained in this work.

**Author contribution**

MP undertook the analysis of the data and the preparation of the manuscript with contributions from all co-authors. TAP and JB contribute to the analysis of wave climate and LR to the analysis of shoreline rates.

*Acknowledgements*

This work is a contribution to the research group RNM-328 of the Andalusian Research Plan (PAI), the project RNM-6547 funded by the Regional Government of Andalusia and the projects GERICO (CGL 2011-25438) and ADACOSTA (CGL2014-53153-R) funded by the Spanish Ministry of Economy and Competitiveness. M. Puig was supported by the FPI grant (BES-2012-053175) associated to the above project. The authors thank Puertos del Estado by providing the wave data.

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





**Table 1**: Summary of the characteristics of each geomorphological unit.

| | Coastal exposure | Beach type | Degree of human development | Artificial structures |
|---|---|---|---|---|
| **Vistahermosa** | Partially exposed | Z-bay | High | Rip-rap revetments<br>Seawalls |
| **Valdelagrana spit barrier** | Protected | Z-bay | High (North)<br>Low (centre and South) | Jetties (North) |
| **Cadiz** | Exposed | Rectilinear | High (North)<br>Low (centre and South) | Seawalls (North)<br>Groynes (North) |
| **Sancti Petri sand spit** | Exposed | Rectilinear | Low | None |





**Table 2**: Photographs available for the study sites

| Year | Type | Nominal scale |
|------|------|---------------|
| 1956 | Aerialphotograph | 1:33,000 |
| 1976 | Aerialphotograph | 1:30,000 |
| 1977 | Aerialphotograph | 1:18,000 |
| 1981 | Aerialphotograph | 1:30,000 |
| 1982 | Aerialphotograph | 1:25,000 |
| 1983 | Aerialphotograph | 1:30,000 |
| 1984 | Aerialphotograph | 1:30,000 |
| 1985 | Aerialphotograph | 1:18,000 |
| 1986 | Aerialphotograph | 1:18,000 |
| 1992 | Aerialphotograph | 1:20,000 |
| 1994 | Aerialphotograph | 1:15,000 |
| 1998 | Ortophotograph | 1 m/pixel |
| 2000 | Aerialphotograph | 1:30,000 |
| 2002 | Ortophotograph | 0.5 m/pixel |
| 2005 | Ortophotograph | 0.7 m/pixel |
| 2007 | Ortophotograph | 1 m/pixel |
| 2008 | Ortophotograph | 0.5 m/pixel |
| 2010 | Ortophotograph | 0.5 m/pixel |



**Table 3**: Shoreline changes in the study area ([a] rate for the period 1982-1984; [b] rate for the period 1985-1992; [c] rate for the period 1956-1977; [d] rate for the period 1981-1984; [e] rate for the period 1984-1986, as certain aerial photographs were not available for these areas).

| Vistahermosa rate of change (m/yr) | | | | Valdelagrana rate of change (m/yr) | | | |
|---|---|---|---|---|---|---|---|
| *Period* | *Bp 1* | *Bp 2* | *Bp 3* | *Period* | *Bp4* | *Bp5* | *Bp6* |
| **1956-1977** | -1.51 | -1.67 | -0.88 | **1956-1976** | 1.45 | 0.57 | -2.59 |
| **1977-1982** | - | 1.23 | 2.26 | **1976-1977** | -6.03 | -5.55 | -31.41 |
| **1982-1983** | - | -1.27 | -4.05 | **1977-1982** | 2.35 | -0.32 | -11.22 |
| **1983-1984** | 0.74[a] | 11.37 | 9.25 | **1982-1983** | - | 23.14 | 9.49 |
| **1984-1985** | -4.13 | -14.70 | -15.77 | **1983-1984** | 16.14 | 15.43 | -10.35 |
| **1985-1992** | 0.36 | 0.86 | 1.76 | **1984-1985** | -14.39 | -7.50 | -27.37 |
| **1992-1994** | -2.91 | 1.23 | 4.48 | **1985-1992** | 11.07 | 1.21 | -10.20 |
| **1994-1998** | -0.60 | -1.71 | -4.38 | **1992-1994** | -7.46 | -16.74 | -26.13 |
| **1998-2000** | 0.07 | 2.39 | 9.80 | **1994-1998** | 0.47 | 2.01 | -12.66 |
| **2000-2002** | -1.77 | -0.52 | -5.29 | **1998-2000** | 4.94 | 7.71 | -0.51 |
| **2002-2005** | -0.87 | -1.62 | -1.22 | **2000-2002** | 4.78 | 0.93 | -12.83 |
| **2005-2007** | -0.13 | 4.17 | 7.48 | **2002-2005** | -2.27 | -3.22 | -5.77 |
| **2007-2008** | -0.36 | -4.39 | -10.58 | **2005-2007** | 7.28 | -3.70 | -0.75 |
| **2008-2010** | 0.52 | -3.31 | -2.94 | **2007-2008** | -7.81 | -15.04 | -14.74 |
| | | | | **2008-2010** | -10.99 | -8.30 | -9.68 |

| Cadiz rate of change (m/yr) | | | Sancti Petri rate of change (m/yr) | |
|---|---|---|---|---|
| *Period* | *Bp7* | *Bp8* | *Period* | *Bp9* |
| **1956-1976** | -0.68 | -1.22 | **1956-1976** | -3.63 |
| **1976-1977** | 11.02 | - | **1976-1977** | -2.66 |
| **1977-1981** | 0.23 | 0.92 | **1977-1981** | -2.24 |
| **1981-1982** | 0.02 | - | **1981-1984** | 8.47 |
| **1982-1983** | -1.61 | - | **1984-1986** | -8.25 |
| **1983-1984** | 1.73 | 4.73 | **1986-1992** | 0.31 |
| **1984-1985** | -14.22 | - | **1992-1994** | -7.68 |
| **1985-1986** | - | -2.67 | **1994-1998** | 3.5 |
| **1986-1992** | 11.96[b] | 2.96 | **1998-2000** | -6.6 |
| **1992-1994** | -9.49 | 4.41 | **2000-2002** | -1.8 |
| **1994-1998** | -0.73 | -1.36 | **2002-2005** | -6.14 |
| **1998-2000** | -3.75 | 2.03 | **2005-2007** | 17.37 |
| **2000-2002** | 1.69 | -7 | **2007-2008** | -9.16 |
| **2002-2005** | -1.99 | 0.00 | **2008-2010** | -5.83 |
| **2005-2007** | 11.04 | 15.52 | | |
| **2007-2008** | -8.52 | -15.23 | | |
| **2008-2010** | -9.49 | -7.65 | | |



**Table 4**: Summary statistics of the correlations between shoreline changes and storm parameters (* Significance levels are < 95%; **Significance levels are < 99%).

| R lineal | Vistahermosa | | | Valdelagrana | | | Cadiz | | Sancti Petri |
|---|---|---|---|---|---|---|---|---|---|
| | **BP1** | **BP2** | **BP3** | **BP4** | **BP5** | **BP6** | **BP7** | **BP8** | **BP9** |
| Mean energy | -0.35 | -0.30 | -0.42 | -0.58* | 0.23 | -0.20 | -0.28 | -0.55* | -0.34 |
| Cumulative energy | -0.03 | -0.47 | -0.51* | -0.58* | -0.30 | -0.16 | -0.51* | -0.59* | -0.23 |
| Peak energy | -0.26 | -0.45 | -0.58* | -0.51 | -0.02 | -0.03 | -0.39 | -0.54* | -0.40 |
| Mean wave power | -0.34 | -0.28 | -0.40 | -0.56* | 0.30 | -0.14 | -0.27 | -0.57* | -0.34 |
| Cumulative wave power | 0.008 | -0.49 | -0.53* | -0.57* | -0.28 | -0.16 | -0.49* | -0.58* | -0.22 |
| Peak wave power | -0.23 | -0.43 | -0.55* | -0.35 | -0.02 | 0.06 | -0.34 | -0.42 | -0.34 |
| Mean erosivity | -0.24 | -0.22 | -0.37 | -0.56* | 0.32 | -0.07 | -0.26 | -0.59* | -0.35 |
| Cumulative erosivity | 0.11 | -0.42 | -0.47 | -0.55* | -0.28 | -0.11 | -0.49* | -0.59* | -0.22 |
| Peak erosivity | -0.08 | -0.32 | -0.50 | -0.33 | 0.001 | 0.12 | -0.32 | -0.53* | -0.37 |
| Mean energy at high tide | -0.44 | -0.33 | -0.43 | -0.59** | 0.23 | -0.15 | -0.24 | -0.54* | -0.29 |
| Cumulative energy at high tide | -0.12 | -0.54* | -0.48 | -0.68** | -0.42 | -0.27 | -0.53* | -0.46 | -0.26 |
| Peak energy at high tide | -0.40 | -0.55* | -0.62* | -0.69** | -0.02 | -0.25 | -0.39 | -0.55* | -0.29 |
| Storm duration | 0.12 | -0.41 | -0.43 | -0.57* | -0.38 | -0.13 | -0.52* | -0.59* | -0.26 |
| Storm frequency | 0.07 | -0.36 | -0.24 | -0.47 | -0.44 | 0.01 | -0.37 | -0.36 | -0.27 |





**Figure 1**. Location map of the study area and the analysed sites I (Vistahermosa), II (Valdelagrana spit barrier), III (Cadiz) and IV (Sancti Petri sand spit).





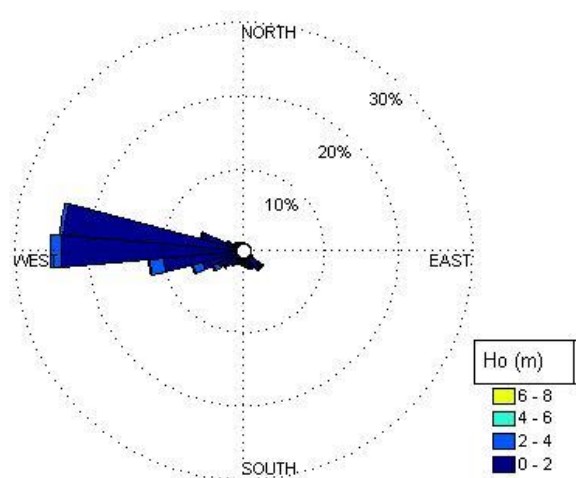

**Figure 2**.Wave height record from 1956 to 2010 in Cadiz buoy.



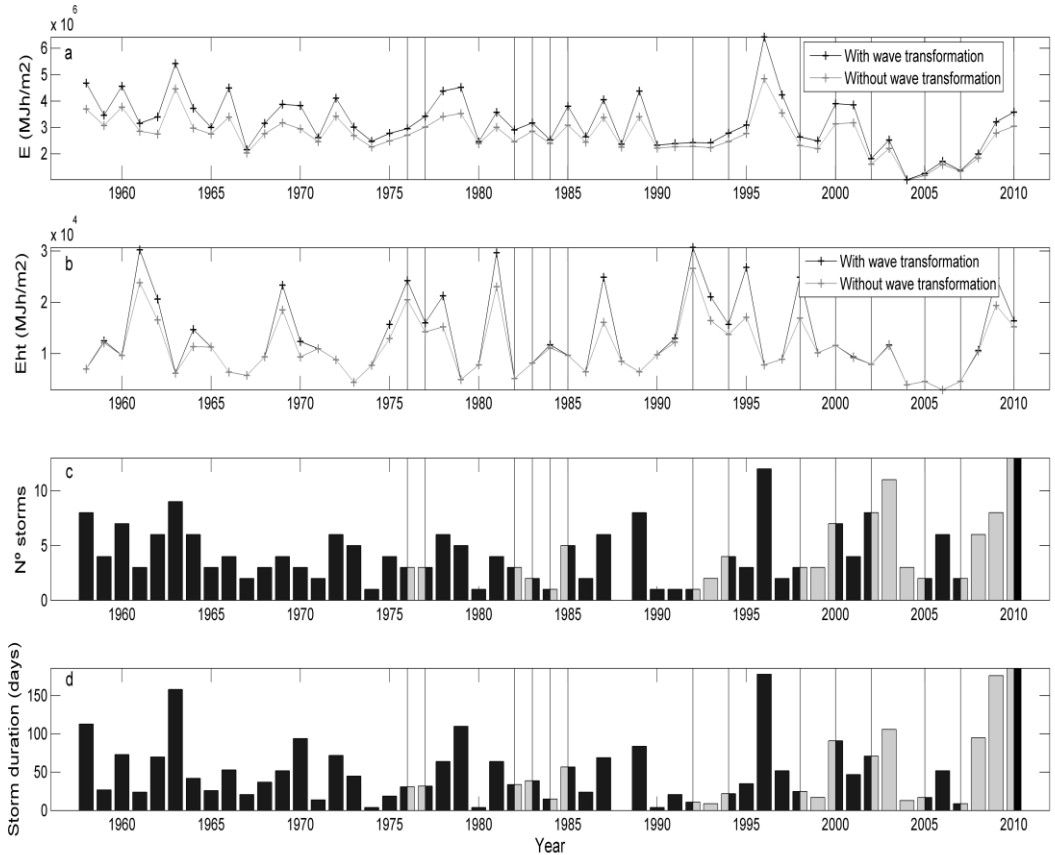

**Figure 3**. Temporal evolution of energetic parameters during the studied period: energy (E), energy at high tide (Eht), number of storms and storm duration. The dates of the available photographs are shown with vertical lines. The effect of wave transformation in E and Eht is shown for Cadiz site.





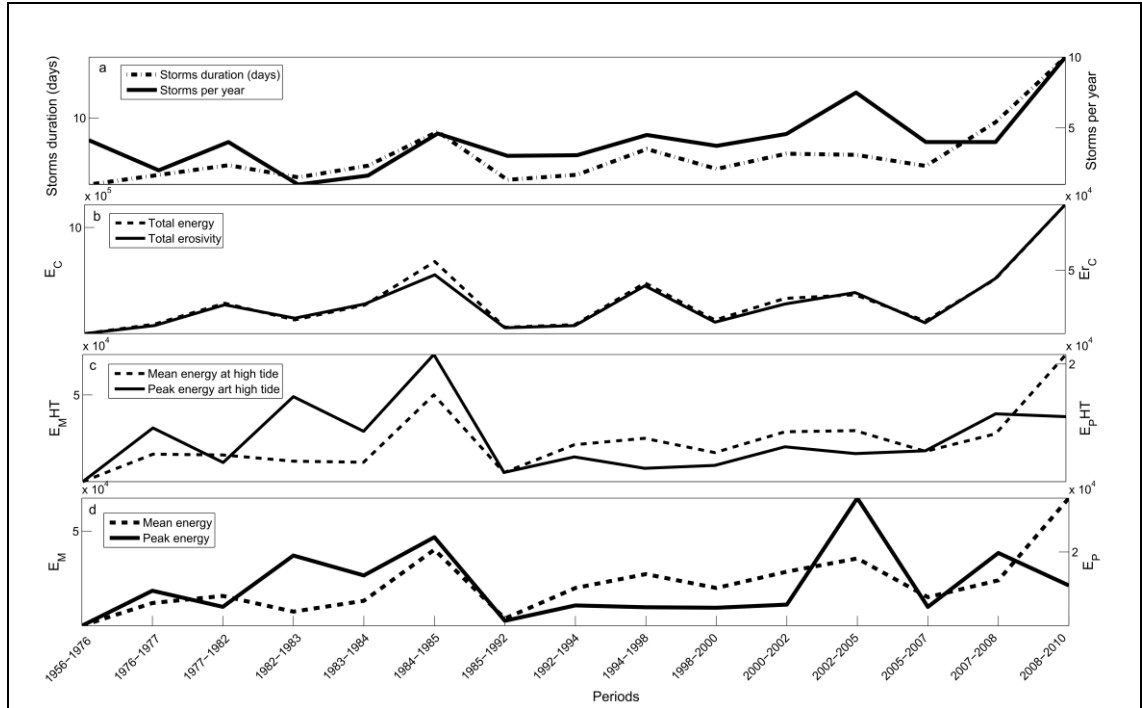

**Figure 4.** Cumulative energy ($E_C$) and wave erosivity ($Er_C$), mean energy ($E_M$) and energy at high tide ($E_M$ht), peak energy ($E_P$) and energy at high tide ($E_P$ht) and storm duration for the periods between sets of photographs.




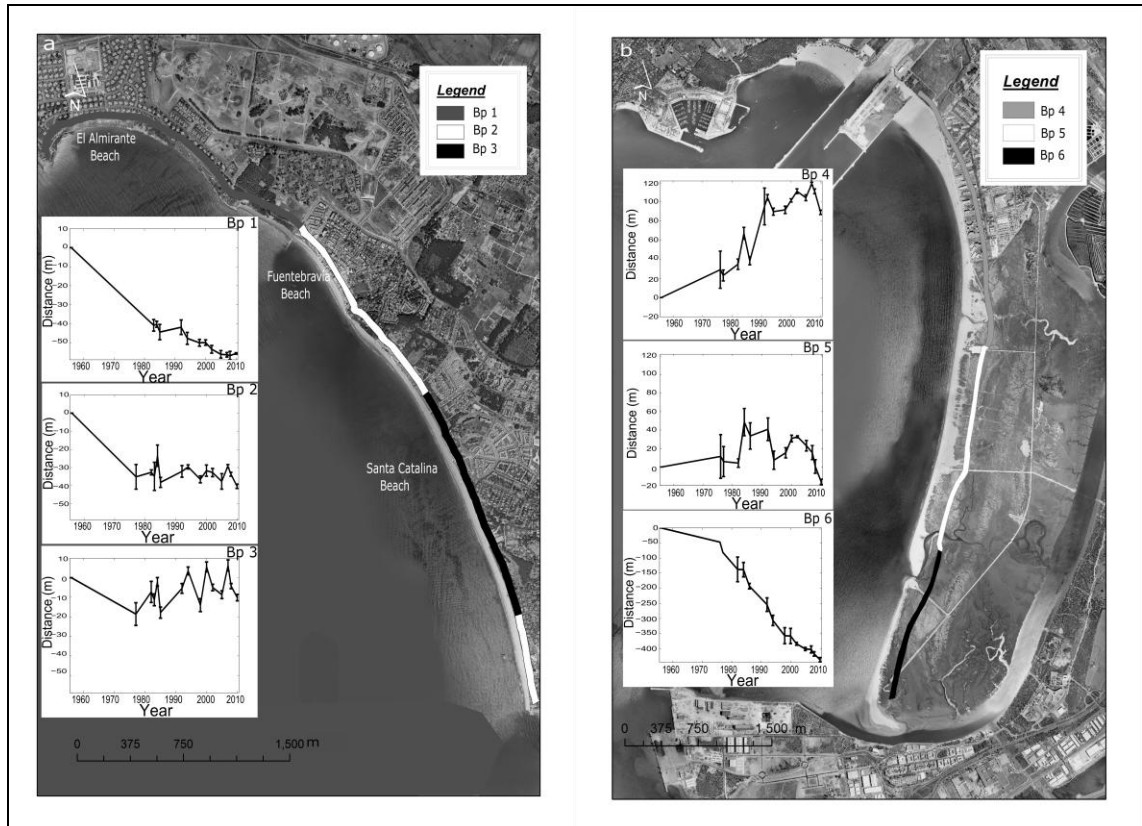

**Figure 5**. Shoreline trends of Vistahermosa (a) and Valdelagrana (b) sites classified by behavioural patterns (BP). Error bars represent alongshore variability.




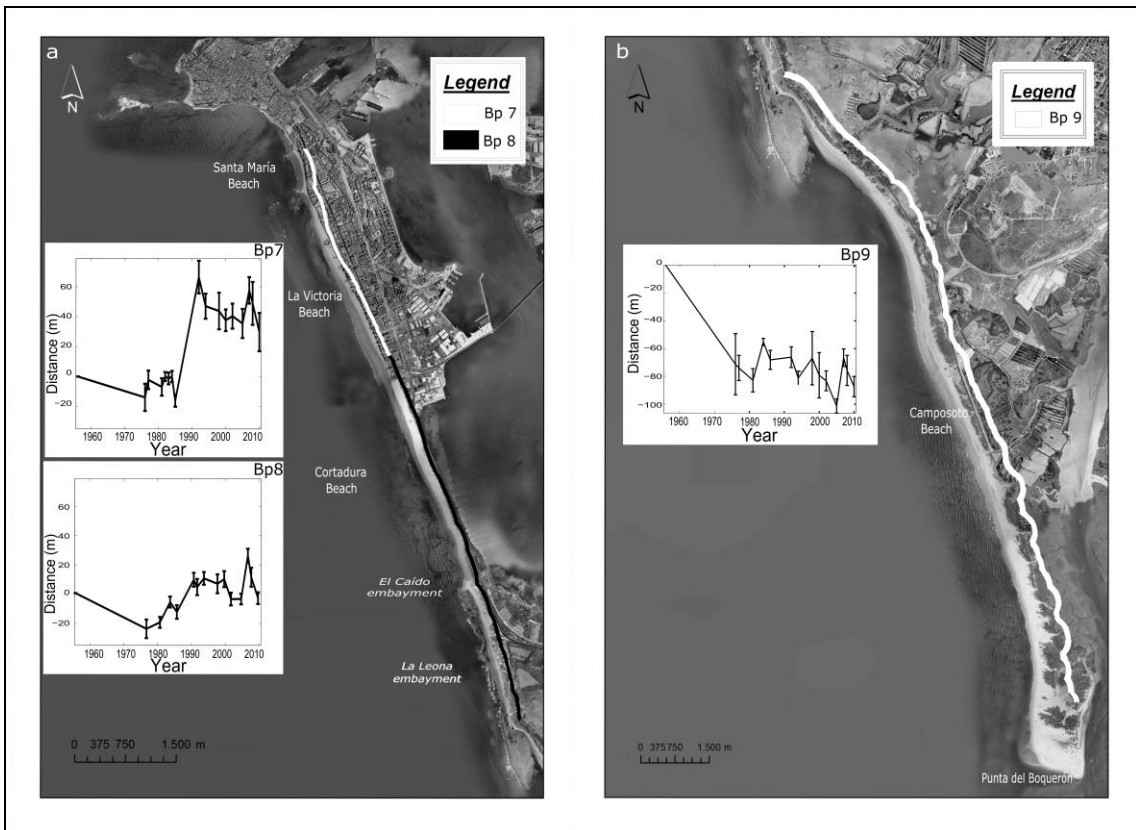

**Figure 6.** Shoreline trends of Cadiz (a) and Sancti Petri (b) sites classified by behavioural patterns (BP). Error bars represent

alongshore variability.




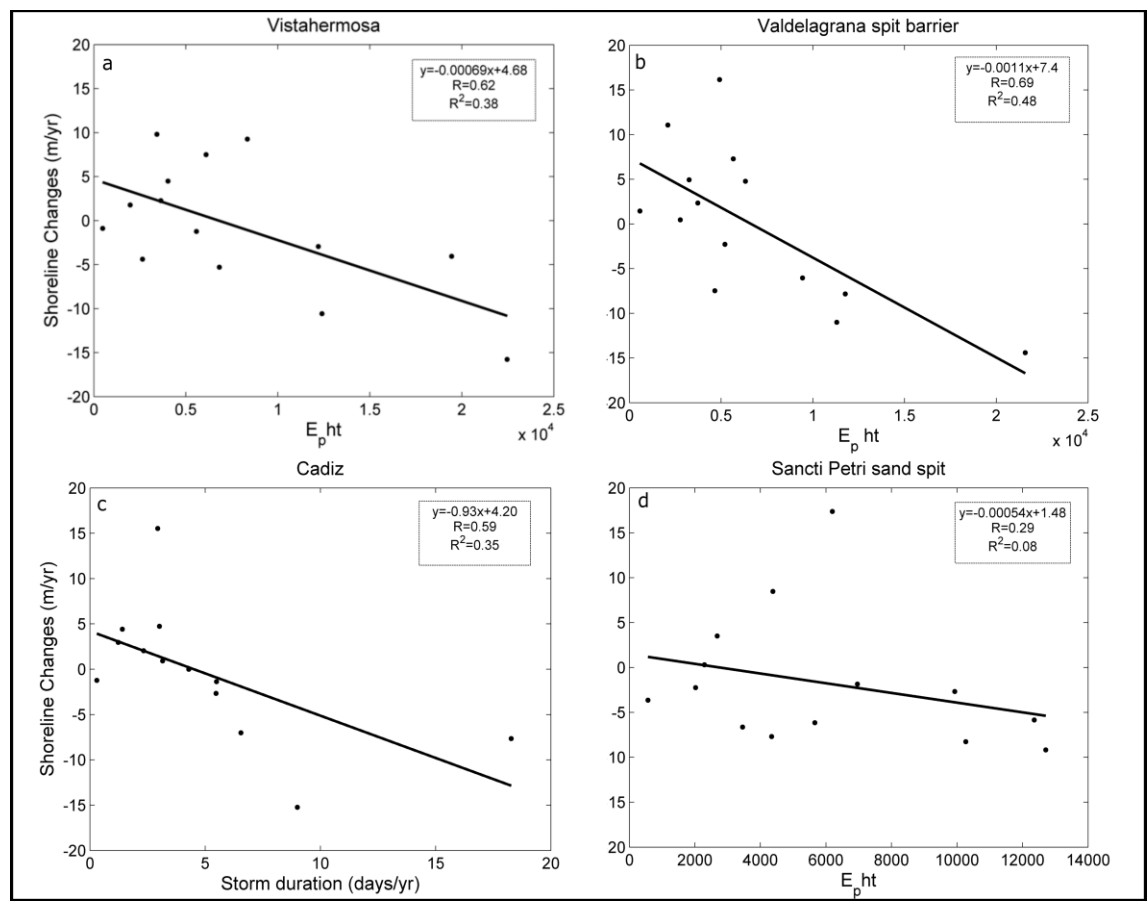

**Figure 7.** Examples of scatter plots of the correlation analysis for the study sites. Peak energy at high tide ($E_p$ht) is shown for

Vistahermosa, Valdelagrana and Sancti Petri units, while storm duration is shown for Cadiz unit.