# Peer review of "Contribution of storms to shoreline changes in mesotidal dissipative beaches. Case study in the Gulf of Cadiz (SW Spain)"

_Natural Hazards and Earth System Sciences, 2016_

## Referee Comment (RC1) · Anonymous Referee #1 · 14 Jul 2016

The paper of Puig et al. analysis in how far storms of different duration and magnitude contribute to medium-term morphological changes along the Gulf of Cadiz in SW Spain. Therefore, the authors combine wave data with remote sensing observations and compare the relationships of the individual components and parameters on a statistical basis. The methods appear sound. They conclude that in highly exposed sub regions within their study area, the observed shoreline changes can be explained by storms whereas in more sheltered areas, this relationship is less obvious and likely masked by anthropogenic activities.

The motivation behind the analysis is comprehensive and important regarding the large uncertainty with respect to future storm occurrences and strengths. The paper is in

most parts well written and I have only a few comments and suggestions for improvements respectively.

Page 3, Lines 11-14: It is difficult to see why the northern part of section 1 is sheltered from storm waves whereas the southern part is exposed. Maybe you can provide some field photographs for this and the remaining sections?

Page 4, Lines 22-23: What are the measurement and hindcast durations for the wave buoy and HIPOCAS respectively?

Page 6, Line 20: A recent publication, which deals with rates of shoreline change and how they are influenced by the geomorphic timescales under consideration, comes from Mann, Bayliss-Smith and Westphal (2016, Journal of Coastal Research). Though they focus on reef islands, the underlying issue is surely the same (see also on Page 14, Lines 4-7).

Page 6, Line 28: Such weighting factors always carry artificial boundaries during the calculation with them as it excludes the detection of a morphodynamics feedback related to earlier storms. However, I acknowledge that this difficulty cannot easily be overcome and I think the present study defines their weighing factors in a comprehensive manner.

Table 2: Please provide the shoreline uncertainties for each data set and how these have been calculated.

Technical comments:

Page 2, Line 10: Explain NOA and EA.

Missing spaces: Page 2, Lines 14, 26 Page 8, Line 4 Page 11, Lines 1, 2, 3
* * *

---

## Referee Comment (RC2) · Anonymous Referee #2 · 14 Jul 2016

This paper examines the relationship between shoreline change and storm parameters along four beaches in the Bay of Cadiz. Shoreline change is measured by digitizing shorelines from historic orthophotographs dating from 1956 to 2010. Storm parameters are calculated from a synthetic storm record based on observed wave data. Correlation between shoreline change and storminess is determined via the Pearson correlation coefficient. The paper is relevant to ongoing coastal hazard studies and for the most part has a sound methodology, although a few things need clarification. However, I think the paper could benefit from an expanded analysis to further explore the mechanisms driving medium-term shoreline change.

Specific comments:

[Figure]

I am a little confused about the temporal scales used in this study. The authors say they are interested in medium-term shoreline change and have data that spans over ∼60 years. However, they calculate shoreline change over periods varying from 1 to 20 years based on the available imagery, which is used to correlate with storm parameters over the same time scales. How did the authors determine the time periods when each shoreline change rate was calculated? How would the results differ if shoreline change rates were calculated over the entire period of record (1956 to 2010) for each Bp and then compared with the storm parameters over the period of record? Likewise, what if the time periods were divided into periods of calm and storminess?

Were temporal outliers identified in the rates of shoreline change for each Bp? If so, what is the source of the outliers? Does removing these outliers improve the correlations?

The main conclusion is that overall there is a low correlation between storm parameters and medium-term (defined here as ∼60 years) shoreline changes; previous studies have also concluded that episodic meteorological forcing doesn't directly influence shoreline change over longer time scales (e.g., Fenster et al., 2001, http://www.jstor.org/stable/4300222). The authors also conclude that anthropogenic influences and geological structure are the main factors impacting shoreline change in the study area. This is something that could be explored further to make new conclusions about the impacting factors of medium-term shoreline change. For example, the authors point out that when tide state is considered, the correlation improves - what about other diurnal processes such as winds and high water levels? Similarly, what about longer-term processes such as sea level rise or changes in sediment supply? This would also provide a better idea of how these results may be applicable to other study areas.

Technical corrections:

P. 2 Line 10: Define NAO and EA.

P.2 Line 14: put space between "assessment" and "is".

P. 5 Equation 5: define Er in the text.

P. 9 Line 8: should be "except" not "excepting".

P. 9 Line 9: I would not call these significant correlations, as the r value is pretty low. There are few more instances where the word significant is used – again I would not use this word.

P. 9 Line 26: put a space between "peak" and "energy".

P. 10 Line 3: should be "except" not "excepting".

P. 11 Line 1: put a space between "described" and "in".

P. 12 Line 11: it would be helpful to include the dates of any anthropogenic features in Table 1.

---

## Author Comment (AC1) · 14 Sep 2016

Dear Referee #1,

The manuscript has been revised according to your suggestions and comments. The following is an item-by-item answer to each comment, which has been pasted in to provide a direct response to each of them.

Page 3, Lines 11-14: It is difficult to see why the northern part of section 1 is sheltered from storm waves whereas the southern part is exposed. Maybe you can provide some field photographs for this and the remaining sections?

The low exposure of the northern part of section 1 (Behavioural Pattern 1) to storm

waves is due to its different coastline orientation (WNW-ESE) with respect to the central and southern parts of the section. This is clearly seen in Figure 1; however, lines 12-13 in Page 3 have been modified, and the following sentence has been added to clarify the content:

The northern part, which presents WNW-ESE orientation (Figure 1), is relatively protected from storm waves as they approach mainly from western-southwestern directions (Figure 2) refracting around Rota headland. We consider that including field photographs would not be convenient, as there are a total of nine behavioural patterns with different extents, so it would be necessary to include a considerable amount of photos. The manuscript already has three figures to present the studied sections: Figure 1 is the location map of the study area and the analysed sites and Figures 6 and 7 are aerial photographs of each section. We strongly believe that these three figures are enough to show the characteristics of the study area.

Page 4, Lines 22-23: What are the measurement and hindcast durations for the wave buoy and HIPOCAS respectively?

The following sentence has been added to show the duration of wave record:

The duration of the data considered in the hindcast database of the HIPOCAS project is between 1958 and 2001, and that of the coastal wave buoy of Cadiz is between 2002 and 2010.

Page 6, Line 20: A recent publication, which deals with rates of shoreline change and how they are influenced by the geomorphic timescales under consideration, comes from Mann, Bayliss-Smith and Westphal (2016, Journal of Coastal Research). Though they focus on reef islands, the underlying issue is surely the same (see also on Page 14, Lines 4-7).

This is an interesting publication that has been added into the revised manuscript to indicate the importance of the temporal perspective in the shoreline change accuracy.

[Figure]

Page 6, Line 28: Such weighting factors always carry artificial boundaries during the calculation with them as it excludes the detection of a morphodynamics feedback related to earlier storms. However, I acknowledge that this difficulty cannot easily be overcome and I think the present study defines their weighing factors in a comprehensive manner.

We completely agree with the referee, and as acknowledged, the inclusion of feedback for earlier storms is out of the scope of this work.

Table 2: Please provide the shoreline uncertainties for each data set and how these have been calculated.

Done

Technical comments:

Page 2, Line 10: Explain NOA and EA. Done

Missing spaces: Page 2, Lines 14, 26 Page 8, Line 4 Page 11, Lines 1, 2, 3. Done

We strongly believe that the above changes have greatly improved the original manuscript, and we hope that the revised manuscript will be suitable for publication in Natural Hazards and Earth System Sciences. Thank you very much.

Yours sincerely,

Maria Puig

———————————————

---

## Author Comment (AC2) · 14 Sep 2016

Dear Referee #2,

The manuscript has been revised according to your suggestions and comments. The following is an item-by-item answer to each comment, which has been pasted in to provide a direct response to each of them.

- I am a little confused about the temporal scales used in this study. The authors say they are interested in medium-term shoreline change and have data that spans over 60 years. However, they calculate shoreline change over periods varying from 1 to 20 years based on the available imagery, which is used to correlate with storm parameters

over the same time scales.

Shoreline variability is affected by natural and human factors that influence at different spatial and temporal scales. The definition of a proper scale is complicated as there are multiple classifications. In the present work, the division proposed by Stive et al. (2002) has been used, which defines the medium term as a temporal scale between years and decades and spatial scale between 1 km and 5 km.

- How did the authors determine the time periods when each shoreline change rate was calculated?

The time periods when shoreline changes were calculated correspond to the time periods between the available aerial photographs. Then the correlation between these shoreline changes (Table 2) and storm parameters for the same periods (i.e. time intervals between consecutive photographs) was investigated.

- How would the results differ if shoreline change rates were calculated over the entire period of record (1956 to 2010) for each Bp and then compared with the storm parameters over the period of record?

We do not consider appropriate the comparison of the shoreline change rate of the entire record with the storm parameters for the same period, as it would yield a single value of shoreline change rate for each zone and hence, it would not be possible to establish a correlation.

-Likewise, what if the time periods were divided into periods of calm and storminess?

The aim of this work is to analyse the contribution of storms to coastal evolution, so in the present analysis a comparison between shoreline changes and storm parameters has been performed. Periods of calm and periods with intense shoreline accretion have been considered as outliers (Page 9, lines 32-33; Page 10, lines 5-6; Page 11, lines 13-14; Page 14, lines 24-26), and as such it would not be appropriate to divide time intervals into periods of calm and storminess and the subsequent analysis.

-Were temporal outliers identified in the rates of shoreline change for each Bp? If so, what is the source of the outliers? Does removing these outliers improve the correlations?

As aforementioned, outliers have been identified in the study area related to strong shoreline accretion and low energy periods (Please see Page 9, lines 32-33; Page 10, lines 5-6; Page 11, lines 13-14; Page 14, lines 24-26 ). The removal of these data has improved the correlations and this is shown along the Discussion and Conclusion sections.

-The main conclusion is that overall there is a low correlation between storm parameters and medium-term (defined here as âĹij60 years) shoreline changes; previous studies have also concluded that episodic meteorological forcing doesn't directly influence shoreline change over longer time scales (e.g., Fenster et al., 2001, http://www.jstor.org/stable/4300222). The authors also conclude that anthropogenic influences and geological structure are the main factors impacting shoreline change in the study area. This is something that could be explored further to make new conclusions about the impacting factors of medium-term shoreline change. For example, the authors point out that when tide state is considered, the correlation improves - what about other diurnal processes such as winds and high water levels?

As stated above, in this work the medium term is defined as years to decades according to Stive et al. (2002), as these are the time intervals between consecutive photographs, hence it is not 60 years (which is the total period analysed). In this scale, natural factors such as wave climate variations and extreme events can induce shoreline changes in the medium term (Stive et al., 2002). The wave height is correlated with storm surge (high water level resulting from low pressure and strong winds) in the study area (Del Río et al., 2012) so it is taken into account in the statistical analysis.

-Similarly, what about longer-term processes such as sea level rise or changes in sediment supply? This would also provide a better idea of how these results may be

applicable to other study areas.

We consider that the effect of sea level rise and river sediment discharges are out of the scope of the present work as they control shoreline evolution on a long-term scale (centuries, millennia) (Cowell and Thom, 1994; Stive et al., 2002). That said, regarding sea level rise, the most reliable estimations in the study area report a rate of 1.0 $\pm$ 0.2 mm yr-1 over the 20th century (Marcos et al., 2011), so the impacts on shoreline changes in the medium term would be negligible when compared to other factors (Del Río et al., 2013). As for changes in sediment supply, the area has experienced general sediment deficit caused by river basin regulation. Although there are studies that relate the reduction of fluvial sediment supply with coastal erosion in the area on a longer term (Del Río et al., 2013), to date the precise contribution of this factor to specific rates of erosion remains unknown.

Technical comments:

P. 2 Line 10: Define NAO and EA. Done

P.2 Line 14: put space between "assessment" and "is". Done

P. 5 Equation 5: define Er in the text. It is defined in Page 5, Line 3.

P. 9 Line 8: should be "except" not "excepting". Done

P. 9 Line 9: I would not call these significant correlations, as the r value is pretty low. There are few more instances where the word significant is used – again I would not use this word.

In this work the use of the word "significant" has been considered appropriate as the Pearson correlation analysis indicates a p-value lower than 0.05.

P. 9 Line 26: put a space between "peak" and "energy". Done

P. 10 Line 3: should be "except" not "excepting". Done

P. 11 Line 1: put a space between "described" and "in". Done

P. 12 Line 11: it would be helpful to include the dates of any anthropogenic features in Table 1. Done

We strongly believe that the above changes have greatly improved the original manuscript, and we hope that the revised manuscript will be suitable for publication in Natural Hazards and Earth System Sciences. Thank you very much.

Yours sincerely,

Maria Puig